# Morphological and Metric Analysis of Medieval Dog Remains from Wolin, Poland

**DOI:** 10.3390/ani15152171

**Published:** 2025-07-23

**Authors:** Piotr Baranowski

**Affiliations:** Department of Animal Anatomy and Zoology, Faculty of Biotechnology and Animal Breeding, West Pomeranian University of Technology in Szczecin, ul. Klemensa Janickiego 33, 71-270 Szczecin, Poland; piotr.baranowski@zut.edu.pl

**Keywords:** archeozoology, body mass estimation, dog, medieval, osteometry, Poland, withers height estimation

## Abstract

This study presents a detailed osteometric analysis of 209 domestic dog remains from early medieval Wolin (Poland), revealing significant variation in body size and morphology. Measurements of skulls and long bones indicate the presence of at least two or three distinct phenotypes, with estimated shoulder heights ranging from 40 to over 60 cm and body masses between 4 and 40 kg. The observed diversity is interpreted in the context of functional roles, trade-related gene flow, and environmental adaptation in a dynamic urban port settlement. The results contribute new data to the study of canine morphology and domestication in the early medieval Baltic region.

## 1. Introduction

### 1.1. Global Context of Dog Domestication

The dog is considered the oldest domestic animal, with remains found from Northern Europe to Palestine [1,2,3,4]. Despite the fact that the dog is the most morphologically diverse mammal—reflected in the cranial and postcranial skeletal proportions, behavioral differences, and physiology [5]—research to date has not been able to unequivocally identify the dog’s direct ancestor [6,7,8]. The most widely accepted view holds that the dog descends from the gray wolf (*Canis lupus*), as supported by multiple research teams [1,2,9,10,11,12]. It is assumed that the differentiation between dogs and wolves began as early as 100,000 BCE [10] or even 125,000 BCE [13], and that individuals with skulls characteristic of the genus *Canis familiaris* date back at least 31,000 years [14]. More recently, morphometric analyses by Galeta et al. [15] have shown that certain European canid skulls dating from 36,000 to 16,500 years ago display features associated with domestication (shorter skulls, broader palates, and braincases), suggesting that the domestication process may have started much earlier than previously believed.

### 1.2. The Human–Dog Relationship in Prehistory and the Early Medieval Period: Utility and Morphology

The earliest discoveries of wolf-like dogs associated with hominid remains, dating to around 400,000 BC [16], suggest an initial, loose form of coexistence between humans and canids. This relationship likely began as a mutualistic exchange—humans offering food scraps and shelter in return for early warning against external threats. Over time, humans came to value not only the companionship of dogs but, above all, their utilitarian potential. Dogs were employed in hunting, herding, guarding, transportation, warfare, and performing sanitary functions [17,18,19]. The Neolithic period marked a significant turning point: the emergence of sedentary economies and changes in lifestyle promoted selective breeding, leading to the appearance of dogs with a wide range of morphological and behavioral traits [20,21]. Variability in size and craniofacial features was already evident at this stage—from small brachycephalic dogs to larger individuals with dolichocephalic skulls. Classical studies by Rütimeyer [22] and Studer [23] describe the so-called “peat dog,” whose remains—recovered from pile deposits around Lake Zurich and the Bernese region—demonstrated that these early domesticated animals were morphologically distinct from wild wolf populations contemporaneous with these dogs. The earliest remains of these dogs, with skull base lengths ranging from 130 to 150 mm, suggest that they were significantly smaller than wolves of the same era. Studer’s work revealed a greater degree of morphological diversity than previously recognized, possibly reflecting early selection for functional traits and environmental adaptability. In Scandinavia, dog burials from Neolithic and Bronze Age contexts, such as those at Skateholm and Jämtland, were frequently accompanied by grave goods, highlighting both emotional significance and symbolic roles within human communities [24]. Comparable patterns have been observed at sites such as Star Carr in Britain and later Iron Age contexts [25,26], where a shift is evident from dogs serving primarily utilitarian roles to being incorporated into ritual- or status-related spheres. The Roman and Late Roman periods witnessed further phenotypic and functional diversification. Morphometric analyses of remains from Pompeii [27] identified distinct types: small brachycephalic lapdogs, medium-sized herding dogs, and large dolichocephalic guard dogs. Iconographic sources such as mosaics and frescoes from Pompeii and other Roman cities, including those outside Europe [28], support this morphological variety and link specific dog types to defined functional roles. Burials from sites such as Fidene, Yasmina (Carthage), and Leicester reveal not only phenotypic differences but also care for dogs exhibiting skeletal pathologies, suggesting emotional bonds between humans and individual animals [29,30,31]. Concurrently, a broader spectrum of cranial morphotypes emerged. Genetic analyses of both modern and ancient dog populations indicate that variability in skull morphology was largely driven by mutations in specific genes, such as BMP3, which influence the proportions and length of cranial elements [32]. These changes have been associated with adaptation to specific tasks, such as vigilance, tracking, or cooperation with humans, facilitated by increased neuroplasticity in selectively bred dogs [21]. In the Middle Ages, human–dog relationships became increasingly diversified. Archaeozoological data from Europe and the Baltic region [33,34] document the presence of dogs of various sizes—ranging from small lapdogs kept in monastic and urban settings to large individuals with dolichocephalic skulls, interpreted as hunting or coursing dogs associated with elite use. Large hunting dogs were integral to courtly culture and served as clear indicators of prestige and high social status. Hunting itself, as a ritualized activity, symbolized dominion over nature and carried considerable ideological significance. In monastic contexts, the breeding of animals—including dogs—was legitimized by their perceived utility in serving human needs. Shifts in dog morphotypes, such as the decline in large hunting breeds, may have been influenced by religious practices, including fasting and the avoidance of meat consumption, which in turn encouraged the maintenance of smaller, more “economical” types. A synthesis of the findings of both authors suggests that medieval dogs functioned not only as utilitarian animals but also as carriers of religious and social meanings. The influence of the Church and the growth of urban centers had a lasting effect on the roles dogs played and on their morphological characteristics, as evidenced in the archeological record. Materials from Lithuania dating from the 12th to 18th centuries [35], as well as from early medieval Anatolia [36] and Western Europe, include dogs displaying traits adapted to specific local roles [37]. The occurrence of individuals with deformities or advanced degenerative changes implies prolonged lifespans and suggests that humans provided care and protection for these animals. Collectively, osteological, iconographic, and textual evidence illustrates the evolution of the dog not merely as a utilitarian asset but increasingly as a social companion, afforded acceptable living conditions and fulfilling diverse roles [28].

### 1.3. Cultural and Ritual Aspects of the Dog’s Presence

Over time, dogs came to be objects of worship [38,39,40], and representations of dogs adorned human-crafted artifacts, often depicting scenes involving dogs [41,42,43]. Evidence for the development of social bonds between humans and dogs is provided by burials of dogs found worldwide; the manner in which these animals were interred testifies to the exceptional relationship between humans and dogs many millennia ago [9,44,45]. However, this reverence did not preclude the consumption of dog meat by humans from the Late Paleolithic through the Iron Age [46,47]. In subsequent historical periods, dogs’ roles in their relationships with humans—relative to other domesticated mammals—increased. Nevertheless, in some medieval and late-medieval communities, dog meat featured in the diet [48], and it remains part of the diet of some people today [25,47,49].

### 1.4. Osteological Materials in Polish Archeology

Mythological sources indicate that dogs in various cultures served as sacrificial offerings in rituals and appeared in religious imagery as attributes of lunar deities within the symbolic life of local communities [44,50]. They were also regarded as companions for the deceased and mediators conveying offerings from the human world to the supernatural realm. The burial of a dog alongside a human may signify the exceptional status of the deceased, with the dog acting as guardian of the grave [3]. The earliest evidence of dogs in the territory of Poland comes from Neolithic settlements, where recovered remains have been taxonomically attributed to *Canis inostrancevi*, *Canis palustris*, and *Canis matris optimae*. These remains date roughly from 4000 BCE to 1700 BCE [51,52,53,54,55].

In Early Medieval faunal assemblages in Poland, dog bones are rare, indicating the animal’s low popularity as edible meat [54,56,57]. Their occasional presence in kitchen refuse likely represents accidental contamination rather than an accurate reflection of population numbers [58,59]. From the Roman period onward into modern times, cultural developments characterized by urbanization—and the accompanying changes in dogs’ roles—were marked by an increasing diversification of dog morphotypes: large individuals declined, while smaller ones became more prevalent. In towns, the popularity of small companion breeds grew, whereas working dogs used for defense, herding, or hunting were probably bred chiefly in rural areas and on manor estates [60]. Over two millennia on the Polish plains, this diversification manifested as a clear shift from the relatively uniform, large dogs of the Roman period to an increasingly varied assortment of smaller breeds in the medieval and modern eras [34].

### 1.5. The History of Wolin: Previous Remains Studies and the Research Gap

Early Settlement: Wolin appears in historical records under various names: Vineta, Iumne, Jom, Jómsborg, and Vuolini [61,62,63,64]. It was an important town in early medieval Europe. The nucleus of Wolin (53°50′35.8″ N; 14°37′06.3″ E) was most likely established at the most convenient crossing point of the Dziwna River [65,66,67]. Its history, spanning from the late 8th century to the end of the 12th century, has been thoroughly described in the literature by its discoverer and foremost scholar of early medieval Wolin, Professor Władysław Filipowiak [68,69,70].

**Figure 1 animals-15-02171-f001:**
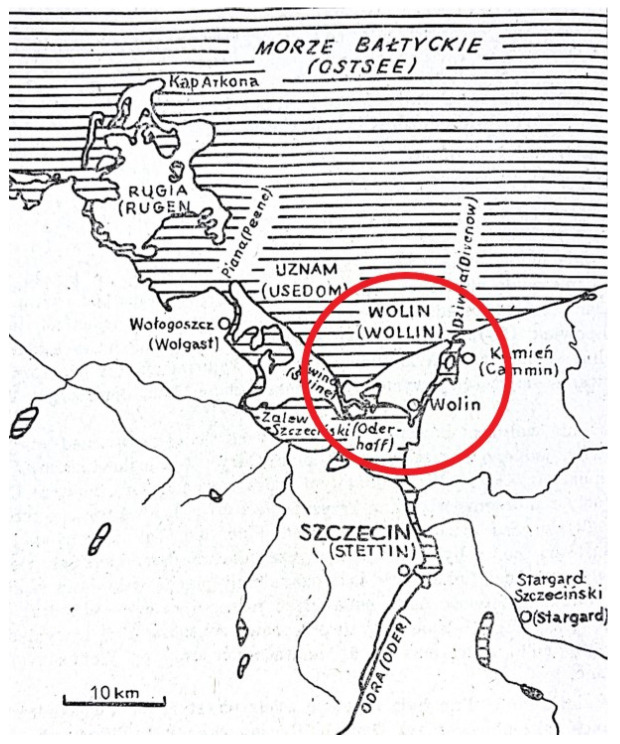
Location of Wolin in relation to the mouth of the Oder River on the southern Baltic coast, modified after Ważny and Eckstein 1987, p. 148, [71].

The primary factor influencing the establishment of the settlement was its strategic location at the crossroads of waterways running west to east across the southern Baltic Sea (Figure 1), serving as a transshipment point for river vessels traveling south into the interior [72]. This advantageous position enabled Wolin to develop into a major center of craft production, trade, and communication in Western Pomerania, as well as an important hub of religious activity [73]. At the time, it was a prosperous settlement, established through the involvement of powerful Pomeranian groups in the lucrative Baltic trade, which enriched the tribal elites of the coastal regions [74]. Wolin Island is part of the Wolin–Uznam archipelago, which bounds the Szczecin Lagoon to the north and lies on the southern shore of the Pomeranian Bay. It is separated from the mainland by the Dziwna River strait and from Uznam Island by the Świna River channel. The entirety of Wolin Island falls within Polish territory and, at 265 km^2^, is the country’s largest island [75]. The island lies in a temperate climate zone with direct influence from the Baltic Sea. The first osteological analysis of animal remains from Wolin was presented by Kubasiewicz in 1959 [76]. In relation to the remains of dogs, the breeding of which was intensified during the period of the greatest development of the manufacturing workshop, the research was focused on determining the forms of this species at that time and especially the size because the early medieval material revealed the existence of a number of breeds in Poland [52,55,60,77,78]. Initial osteometric and anatomical assessments indicated the presence of three main types in early medieval Wolin: (1) a form resembling the so-called pile-dwelling dog (*Canis palustris*); (2) medium-sized dogs of the *Canis intermedius* and *Canis matris optimae* types; and (3) large dogs of the *Canis inostrancevi* type. These were not pure breeds, but likely hybrids resulting from repeated crossbreeding. Studies have shown that medium-sized and smaller dogs were the most common, likely fulfilling roles related to guarding, herding, and/or hunting [79].

Although previous research identified the basic dog morphotypes and size categories in Wolin, a comprehensive, systematic osteometric, and morphological analysis of the fragmentary remains—combined with a contextual interpretation of their functional roles within the city’s social and economic structure—has yet to be undertaken.

### 1.6. Research Objectives

The objectives of this study are to conduct a comprehensive osteometric and morphological analysis of canine remains from early medieval Wolin, with particular emphasis on

Examining variability in linear dimensions of cranial and long bones and estimating withers’ height and body mass;Determining the range of morphotypic variation based on fragmentary material;Interpreting the possible functions these animals served within the Wolin community;Comparing the results with data from other areas of Pomerania.

### 1.7. Archeological Context

The material examined in this study comprises domestic dog bone remains. These were isolated from a curated assemblage of mammalian skeletal elements and fragments, recovered between 1956 and 2003 from cultural strata dated to the late 6th through the early 13th centuries AD. The assemblage derives from six excavation trenches at the Wolin archeological site (Figure 2), excavated by the Archeological Station of the Polish Academy of Sciences’ Institute of Material Culture History in Wolin, with subsequent osteological analysis conducted by researchers from the Department of Animal Anatomy in Szczecin [76,80,81,82]. Contextual dating spans from circa the late 8th century AD [up to 850 AD [83] to the early 13th century AD [79,84].

Excavations focused on several discrete loci within the early medieval urban complex on the left bank of the Dziwna River: Wolin–Miasto, interpreted as a commercial, craft-production, and port district active from the 9th to the 13th centuries AD, and Wolin–Srebrne Wzgórze, a defensive “production-type” settlement dating to the 11–13th centuries AD.

### 1.8. Dating

Dendrochronological analysis of port remains at Wolin dates the quay’s origin to 880–896, with gradual expansion through 994. Wood from trees felled for the construction of the pagan temple dates to the turn of 965–966. Dating of the Silver Hill craftsmen’s quarter—located north of the former settlement—indicates building activity as early as 838, with artisanal development continuing into the 10th and 11th centuries AD. Samples from marketplace excavations yield absolute dates at the end of the 9th century [71,85], which broadly correspond with radiocarbon dating [83,86].

## 2. Materials and Methods

The osteological material for this study was recovered from early Medieval cultural strata at two sites in Wolin—Wolin Town and the adjacent Silver Hill. Skeletal assemblages were selected from the faunal remains that had been identified, curated, and stored by the Department of Animal Anatomy and Zoology in Szczecin. In total, 209 dog bones were osteometrically measured, comprising 13 skull, 47 mandibles, 21 scapulae, 33 humeri, 34 radii, 18 ulnae, 16 femora, and 27 tibiae. Each specimen received a unique excavation number and the corresponding site and/or trench code upon recovery. These provenience data were then used to establish the assemblage’s temporal framework. Bone counts by context and their associated dates are presented in Table 1.

Age and Sex of the Dogs: Anatomical terminology for identifying individual bones and their constituent elements follows standard osteometric references. Age was estimated from the eruption and wear of maxillary and mandibular teeth, using the tables and criteria of Habermehl [87] and Lutnicki [88]. Due to extensive tooth loss in the maxillae and mandibles of these early medieval dogs; pronounced wear on the I, M1, and M2 surfaces; obliterated cranial sutures; and the isolated condition of many skeletal elements, all individuals were classified as adults at death. Sex determination was based on morphological differences in the basioccipital region and the spacing of the tympanic-bulla fissures, following the classification by The and Truoth [89] and Trouth et al. [90]. Because no reliable method exists for sexing isolated post-cranial elements [91], no attempt was made to control for sex-based variation in the post-cranial measurements.

Measurement Protocol: Both the complete bones and measurable fragments of crania and long bones were documented. Measurements were performed in triplicate using digital and articulated (“arc”) calipers to the nearest 0.01 cm, according to von den Driesch 1976 [92]. The mean of the three readings was calculated, with measurement error ranging from 1% to 3%.

Statistical Analysis and Calculation Formulas: All data were entered into Statistica v. PL and used for all computations. The morphological completeness of the assemblage permitted the estimation of withers height from measurements of appendicular-skeleton segments: the pectoral girdle (*zonopodium anterius*)—*scapula* (n = 5); the *stylopodium* of the forelimb—*humerus* (n = 33); the *zeugopodium* of the forelimb—*radius* (n = 19) and *ulna* (n = 12); the *stylopodium* of the hindlimb—*femur* (n = 13); and the *zeugopodium* of the hindlimb—*tibia* (n = 21). Height was calculated using the regression equations proposed for each segment by Harcourt [25] and the coefficients provided by Lasota-Moskalewska [93].

Cranial-cavity measurements (*basion–ethmoideum*) were applied using the formula by Wyrost and Kucharczyk [78], which was later validated for dog-height estimation by Chrószcz et al. [94] and modified for other taxa [95,96]:H = 1.016 × D − 31.2
where H is withers height in cm and D is the internal *basion–ethmoideum* length of the *neurocranium* in mm.

Body mass was estimated using Onar’s [36] formulas based on the circumferences of the *humerus* and *femur* shafts:Weight (g) = 10^(2.47×log(h)−2.72)^Weight (g) = 10^(2.88×log(f)−3.40)^
where log(h) is the circumference of the *humerus* measured at 35% from the distal end and log(f) is the circumference of the *femur* measured at its midshaft.

Additionally, two mandible-based formulas including mandibular height at M1 (mm) and *femur* circumference (mm) were applied [97]:1. log y = 2.1122 × log x + 1.27222. log y = 2.2574 × log x + 1.1164
where log y = body mass (g) and log x = mandibular height at M1 (mm).3. Weight (g) = 0.35 × √(femur circumference)/1.50

The area of the *foramen magnum* was measured using MultiScanBase v. 18.03 with the ScanBase image–text database. Each cranium was mounted vertically, rostrum down, beneath a calibrated Sony Alpha-100 camera (Sony Corporation, Tokyo, Japan) with a Tamron SP Di-AF 90 mm 1:2.8 MACRO 1:1 lens (Amron Co., Ltd., Saitama, Japan), ensuring that the plane of the foramen magnum was horizontal and perpendicular to the optical axis [98].

In total, 100 metric variables were recorded (46 cranial, 20 mandibular, and 34 post-cranial), from which 37 cranial indices and 6 mandibular indices were derived. Withers height and body mass were estimated using 17 different formulas [36,48,78,93,97].

Sex determination followed Trouth et al. [90] in the absence of the *baculum*. Measurements were taken in the *basioccipital* region between the base and the line connecting the midpoints of both *tympanoccipital* fissures: width (W) is the distance between the lateral-most points of the fissures; length (L) is the distance from the base to the midpoint of the line joining their medial-most points. The sex index (S) is calculated asS = W × 100/LS < 123: maleS > 136: female123 ≤ S ≤ 136: juvenile or castrated individual

## 3. Results

Overall, 51.67% of the analyzed canine skeletal material derives from contexts dated to the late 9th through the mid-11th centuries AD. A further 23.08% originates from layers dated to the second half of the 12th through the third quarter of the 13th centuries AD (Table 1). From this later phase, ten cranial elements (four crania and six mandibulae) and 38 post-cranial specimens were recovered. Material dating to the end of the 8th century to ca. 850 AD and to the second half of the 9th century AD constitutes only 3.36% of the assemblage (three crania and four mandibulae); however, all three crania are sufficiently well preserved to permit detailed craniological analysis.

### 3.1. Sex

No complete skeletons preserving the baculum were recovered; therefore, sex determination of the cranial specimens followed the method by Trouth et al. [90]. The sex index (S) was calculated as the ratio of the inter-tympanoccipital fissure width at its most lateral points (W) to the distance from the basion to the midpoint of the line connecting the most medial points of the tympanoccipital fissures (L) and then multiplied by 100 (S = W × 100/L). The mean S value was 103.69 ± 11.9 (range 82.31–116.58) for male crania and 143.78 ± 5.51 (range 139.14–150.88) for female crania—confirming that S < 123 indicates a male cranium, whereas S > 136 indicates a female cranium. Following Brassard and Callou’s [99] caution regarding historical specimens, S values for the 9th–13th-century AD crania were compared with those from a reference series of 13 modern dog crania of known sex (breeds included the Polish Tatra Sheepdog, the German Shepherd, the Great Dane, the Alaskan Malamute, the Dachshund, and mixed breeds). The concordance of indices (Table 2) confirms the reliability of this method.

### 3.2. Age

All surviving teeth remaining in the alveoli represent permanent dentition. Three of the thirteen male crania are edentulous, while four others retain only a single caninus that is heavily worn. In the maxillae, P^2^–M^2^ teeth are still present but exhibit advanced occlusal wear, consistent with adult age. One male skull lacks the M^1^ alveolus entirely (Figure 3).

To supplement dental aging—given taphonomic and dietary wear effects—we assessed the closure of four synchondroses (*synchondrosis intersphenoidalis*, *synchondrosis sphenooccipitalis*, *synchondrosis interoccipitalis basilateralis*, and *synchondrosis interoccipitalis squamosolateralis*) and nine suturae (*sutura frontoparietalis*, *sutura interparietalis*, *sutura occipitoparietalis*, *sutura interfrontalis*, *sutura frontomaxillaris*, *sutura zygomaticomaxillaris*, *sutura temporozygomatica*, *sutura lacrimomaxillaris*, and *sutura lacrimozygomatica*) [100,101]. Five crania were aged 3–5 years, while six were 5–7 years, and two were 7–10 years at death—a range reflecting ecological and allometric influences on suture closure timing [102,103].

### 3.3. Evidence of Trauma or Damage to Skeletal Material

Two crania exhibit bone lesions (Figure 4).

The first, from a dog aged 3–5 years (No. 2947 A955/10), measures 11.65 mm × 8.10 mm and is located on the margo supraorbitalis sinistra of the os frontale, between the origin of the processus zygomaticus ossis frontalis and the course of the linea temporalis sinistra. The second lesion, from an individual aged 5–7 years (no. 831/11), measures 15.61 mm × 7.62 mm and occurs at the junction of the sutura frontomaxillaris and the sutura frontolacrimalis—sites typically impacted by blows aimed at the supraorbital and facial regions to restrain aggressive behavior [104]. Neither fracture appears to have been fatal or intended as a killing blow [48]. Other damage, or the absence of bone and dental elements, likely results from burial taphonomy—soil movement, layer mixing, and decomposition processes. Damage is most pronounced in the splanchnocranium and less severe in the neurocranium; however, in all cases sufficient anatomy remains to permit the majority of osteometric measurements.

### 3.4. Osteometric Features of the Cranium

A total of 46 measurements were taken on the skulls (Table 3). To calculate the dimensions of certain missing structures, bilateral anatomical symmetry was used. The mean length of eight osteologically mature dog skulls is 197.25 mm ± 14.95. The facial part of the skull—which in modern breeds is highly variable—is slightly to moderately concave in the specimens studied. The mean value of the frontal position index (P-So × 100/So-O) was 120.47 ± 8.83. Among the specimens examined, three skulls have an index value above 125 (x = 128.72 ± 1.76) (medium-snout dogs), while the remaining five exhibit values ranging from 104.25 to 121.74, classifying most as short-snout dogs [105].

In lateral view, a well-developed sagittal crest (*crista sagittalis externa*) and nuchal crest (*crista nuchae*) are clearly visible on 10 of the 13 skulls. The mean height of the sagittal crest above the dorsal parietal surface is 6.98 mm (see Table 3: item 38 minus item 39). However, it should be noted that on four of the skulls, the sagittal crest is only faintly raised above the parietal bone surface, which makes the mean crest height on the remaining specimens (n = 9) 7.87 mm ± 4.09 (3.57 mm–15.61 mm). These four skulls with only a trace sagittal crest date from the second half of the 11th century to the early 13th century AD.

In order to obtain measures that jointly represent the analyzed skulls—and that can serve for simplified analysis via the aggregation of data from the measurement sources—37 cranial indices were estimated (Table 4).

The internal length of the cranial cavity (*Basion–Ethmoideum*) ranges from 78.19 to 87.70 mm. Based on this measurement, the value of the Wyrost–Kucharczyk coefficient [78]—one of those used to estimate withers height—indicates that these are skulls from dogs with an estimated withers height of approximately 48 to 57 cm. The tallest individual, with an estimated withers height of 66.64 cm, was probably a male dating to the period 1000–1050 AD (Figure 5).

The correlation of age, sex, and withers height (calculated using the Wyrost–Kucharczyk index) showed that among dogs aged 3 to 5 years, five skulls correspond to withers heights from 45.74 to 56.46 cm. In the 5- to 7-year age range, three females had estimated withers heights of 52.60 to 57.98 cm, and three males ranged from 52.45 to 66.64 cm. Skulls belonging to two males aged between 7 and 10 years indicate withers heights of 47.85 and 51.81 cm (Table 8).

### 3.5. Osteometric Features of the Mandibula

For the measurements of dog mandibles from Wolin, specimens that represented complete or nearly complete mandibles and allowed for the collection of principal measurements (Table 5) and the calculation of indices (Table 6) were selected. The greatest mean length of osteologically mature mandibles was 136.20 mm ± 11.08.

Using formulas applied to estimate the body weight of dogs [36,97] based on the height of the mandible at the insertion of the M1 tooth, the mean weight of the Wolin dogs was determined to be either 9.82 kg ± 2.14 or 15.10 kg ± 3.48, depending on the mathematical model used (Table 8). In both cases, the coefficients of variation were 21.77% and 23.09%, respectively, indicating considerable variability in body weight.

Based on mandibular length and using a formula for estimating withers height [106], the specimens predominantly fall into the medium and medium–large size categories (Figure 6).

### 3.6. Osteometric Features of the Pectoral Limb Bones

#### 3.6.1. Scapula

The estimated mean height of the scapula for 5 of the 21 specimens in the assemblage was 126.94 mm ± 20.05 (Table 7). The preservation of the material allowed for the estimation of the minimum length of the scapular neck, the maximum length of the glenoid process, as well as the length and width of the glenoid cavity. The coracoid process (*processus coracoideus*) and the glenoid cavity (*cavitas glenoidalis*), despite minor edge damage, were sufficiently well-preserved to permit measurement. Using scapular height (tl) and the formula scapula H = tl × 4.06, the mean withers height was estimated at 51.54 cm ± 8.14 (Table 8).

#### 3.6.2. Humerus

Of the 33 specimens of this bone, 15 were suitable for measuring its maximum length. The mean GL value was 152.31 mm ± 28.33 (Table 7). In the remaining specimens, damage to the greater tubercle (*tuberculum majus*) and/or the lesser tubercle (*tuberculum minus*) at the proximal end prevented the measurement of maximum and lateral lengths. The mean difference between the lateral and maximum lengths of the humerus was 5.76 mm ± 2.00. For specimens where GL measurement was possible, the breadth-to-length index (SD × 100/GL) and the robustness index (CD × 100/GL) were calculated, with mean values of 8.35 ± 1.08 and 26.90 ± 3.21, respectively.

Estimated withers heights based on humeral length ranged from 49.58 cm ± 9.71 to 51.33 cm ± 9.54, with the tallest individuals reaching 62–63 cm. These specimens date to the late 9th century and the transition from the 10th to the 11th century AD (Figure 7).

#### 3.6.3. Radius and Ulna

The greatest mean length of the radius was 146.31 mm ± 37.40 (Table 7), with a coefficient of variation of 25.56%. A similar mean length was observed for the ulna, at 146.34 mm ± 34.56, with a CV of 23.61%. These measurements indicate considerable variability in the dogs’ heights. Estimated withers heights—calculated using the formulas by Harcourt [25] and Lasota-Moskalewska [93]—ranged from 47.11 cm ± 12.04 to 48.47 cm ± 11.89 for the radius and from 39.07 cm ± 9.23 to 41.30 cm ± 9.61 for the ulna (Table 8).

**Table 8 animals-15-02171-t008:** Estimates of shoulder height (cm) at the withers and body mass (kg) of dogs found on Wolin Island from the late 8th century to the third quarter of the 13th century AD.

The Height (H) of the Dog Calculated According to	Statistical Values				
	n	x ± s	Min	Max	V%
Wyrost i Kucharczyk [78]					
the cranial cavity H = 1.016 × D − 31.2	13	53.57 ± 5.39	45.74	66.64	10.12
Mandible height at M_1_ (mm) according to Clark [97]					
1. weight (kg)	31	9.82 ± 2.14	4.38	14.36	21.77
2. weight (kg)	31	15.10 ± 3.48	6.35	22.63	23.09
Harcourt [25]					
humerus H = (3.43 × tl) − 26.54	15	49.58 ± 9.71	31.64	62.10	19.59
radius H = (3.18 × tl) +19.51	19	48.47 ± 11.89	33.36	65.50	24.53
ulna H = (2.78 × tl) + 6.21	12	41.30 ± 9.61	27.39	62.06	23.26
femur H = (3.14 × tl) − 12.96	13	50.52 ± 11.32	29.79	65.90	22.42
tibia H = (2.92 × tl) + 9.41	21	47.34 ± 10.71	27.15	61.32	22.62
Lasota-Moskalewska [93]					
scapula H = tl × 4.06	5	51.54 ± 8.14	37.63	58.46	15.79
humerus H = tl × 3.37	15	51.33 ± 9.54	33.70	63.62	18.59
radius H = tl × 3.22	19	47.11 ± 12.04	29.78	65.36	25.56
ulna H = tl × 2.67	12	39.07 ± 9.23	25.71	59.01	23.61
femur H = tl × 3.01	13	49.67 ± 10.86	29.80	64.41	21.85
tibia H = tl × 2.92	21	46.41 ± 10.71	27.15	61.32	23.08
Femur circumference (mm) according to Clark [97]					
3. Weight (kg)	16	26.84 ± 8.84	9.09	41.15	32.93
Weight estimation (kg) according to Onar [36]					
W = 10^(2.47×log (h))−2.27^	28	19.13 ± 6.08	6.84	33.00	31.76
W = 10^(2.88×log (f))−3.4^	16	17.45 ± 5.78	5.86	26.84	33.13

Explanations: tl—total length; 1. Log y = 2.1122 (log x) + 1.2722, Log y = weight (g), Log x = mandible height at M_1_ (mm). 2. Log y = 2.2574 (log x) + 1.1164, 3. Weight (g) = ^0.35^√femur circumference (mm)/1,50, ^(h)^—humerus; ^(f)^—femur.

Radius lengths indicate that the tallest individuals date to the late 11th century through the second half of the 12th century AD (Figure 8), whereas ulna lengths suggest that the tallest dogs are from the turn of the 10–11th centuries or the first half of the 11th century AD (Figure 9).

**Figure 8 animals-15-02171-f008:**
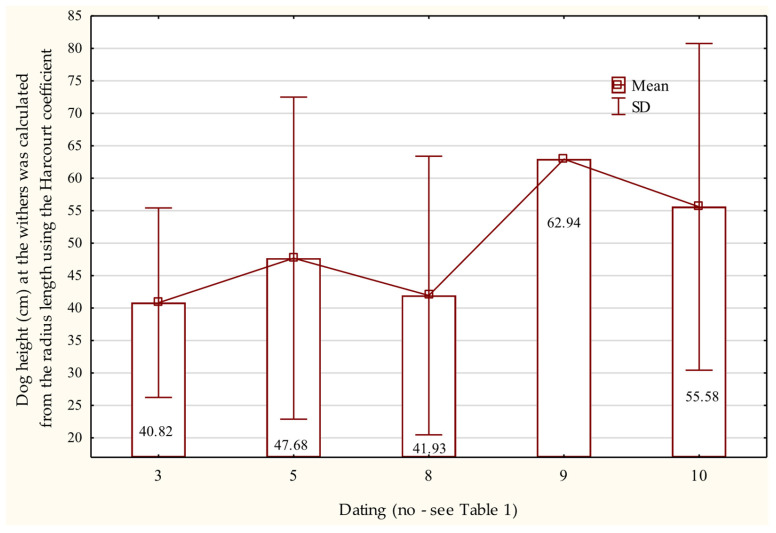
Height at the withers of dogs based on the length of the radius bone in the Early Middle Ages.

**Figure 9 animals-15-02171-f009:**
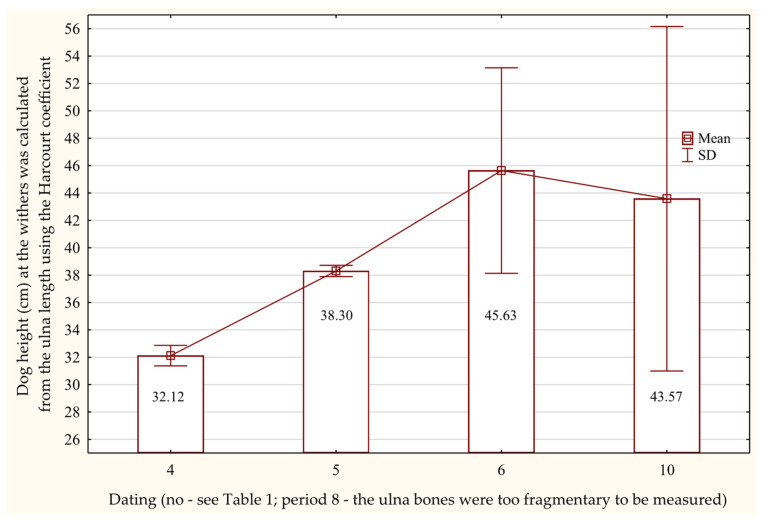
Height at the withers of dogs based on the length of the ulna bone in the Early Middle Ages.

### 3.7. Osteometric Features of the Pelvic Limb Bones

#### Femur and Tibia

The femurs found in the analyzed assemblage are characterized by the greatest mean length, amounting to 165.01 mm ± 36.07, with minimum and maximum recorded values of 99.00 mm and 214.00 mm, respectively (Table 7). The mean midshaft circumference of the femur is 40.30 mm ± 5.41, and the coefficient of variation for this trait is 13.42%, indicating low variability within the sample. The measured greatest length of the tibiae shows a mean value of 158.93 mm ± 36.68. The values of the greatest length (GL) of each measured femur and tibia used to estimate the average withers height of dogs from Wolin indicate that the animals ranged in height from 46.41 cm ± 10.71 to 50.52 cm ± 11.32 (Table 8). Two femurs dating to the turn of the 10th and 11th centuries AD show measurements corresponding to a withers height of 60.0 to 65.90 cm (Figure 10).

Body mass calculated based on the dimensions of both bones suggests that the heaviest individuals could have weighed between 20 and 25 kg, with one exceptional case reaching 33 kg (Table 8). Estimates based on the circumference of the humerus and femur place the dogs’ body mass in the range of 5.86 kg to 41.15 kg, with mean values ranging from 17.45 kg to 26.84 kg. In turn, the greatest length of a tibia from dogs dated to the second half of the 9th century AD and the turn of the 9th and 10th centuries AD indicates that they belonged to animals with an average withers height of 52.43 cm ± 8.55 (Figure 11).

## 4. Discussion

The results indicate that the dog population in Wolin exhibited markedly diverse body sizes as well as cranial and limb morphologies. Small individuals—with estimated body masses of 4–6 kg, depending on the estimation method—and robust forms weighing up to 40 kg are represented (Table 8). Such wide variation in size and morphotype may reflect differing roles of these animals in medieval Wolin. However, their precise functions—similar to interpretations of dog burials accompanied by grave goods [44,50,107]—cannot be firmly established due to insufficient contextual data. This raises the question of whether these dogs were subject to controlled breeding and artificial selection to create distinct breeds, or whether only the most capable working animals were bred, while others lacking desired traits were culled—effectively producing mongrel populations.

Given the pronounced polymorphism of dogs, it is difficult to infer a specific breed standard or phenotypic type from the assemblage, or to relate it to modern breed templates [108,109]. Environmental diversity and diets integrate morphotypes into the local ecosystem [35], shaping canine body forms. Thus, comparisons with modern breeds may be misleading. Notably, the diversity of dogs in Wolin appears greater than in many other West Pomeranian sites (Table 9).

For example, well-preserved adult specimens from Kołobrzeg [81] include crania with *basion–sphenoidale* lengths of 42 mm, *fronto-nasal* lengths (Br–N) of 49–51.5 mm, maximum frontal breadths (Ect–Ect) of 38–49 mm, and cranial vault widths (Eu–Eu) of 51–59 mm. Mandibular dimensions (Goc–Id) ranged from 118 to 136 mm; corpus height anterior to P^3^: 18–20 mm; ramus height (Gov–Cr): 49–52.5 mm; and maximum corpus depth: 10.5–11 mm. Humeral lengths were 146–151 mm, and a single radius measured 118 mm. The authors concluded that the Kołobrzeg dogs were chiefly medium-sized (*Canis intermedius*), with some slightly larger than the palisade-spitz type (*Canis palustris*). At the early medieval gord in Mścięcino, scapulae and long bone fragments matched both medium and smaller Pomeranian forms [112]. In Stargard’s suburbium, measurements of cranial vault width (Eu–Eu: 60.5 mm), *processus mastoideus* breadth (37.8 mm), *foramen magnum* width (21.6 mm), and occipital triangle height (A–B: 42.5 mm) indicated a large individual [113]. At the Dobra Nowogardzka castle, dog remains ranged from small to medium-sized, with some exhibiting angular limb deformities [110].

The comparison of early medieval assemblages from Szczecin’s vegetable market [111] with those from Wolin and Kołobrzeg reveals dogs of medium and large sizes, with some individuals reaching 65 cm at the withers (Table 9). Mandibular lengths (Goc–If: 138–143.5 mm), corpus heights anterior to P^3^ (17–21 mm), and corpus depths (11–12 mm) align with medium/large forms (*Canis intermedius* and *Canis matris optimae*) and small, spitz-derived types (*Canis palustris*).

Thus, both the Wolin data and the comparative literature from West Pomeranian sites demonstrate similar height ranges and likely body-type categories. Compared with dogs of earlier periods—such as specimens from the Migration Era [114] or those dating from the 10–12th centuries AD [115,116]—Wolin dogs were generally smaller.

Recent advances in AMS radiocarbon dating, isotopic analysis, and ancient DNA studies have revealed substantial genetic and morphological diversity within urban dog populations, promoting gene flow among regions [35,117,118]. This supports the likelihood that Wolin’s canine gene pool was similarly replenished or admixed—especially considering evidence for Scandinavian-derived dogs in Pomerania [108]. The variation in size and morphotype among Wolin dogs can thus be seen not only as an adaptation to environmental change, including the evolving resource base of a growing town that shaped living conditions for dogs, but also as a result of human cultural and economic preferences [119,120]. In port cities such as Wolin (and Gdańsk [34]), the importation of foreign stock and admixture with local dogs would have fostered hybrid morphotypes—an interpretation supported by genetic studies from the Baltic region [121,122].

Contemporary craniological studies show that, following detailed morphometric analyses, researchers typically classify dog skulls into defined morphotypes. In modern dogs, these morphotypes are often associated with specific breeds by cynologists. However, applying this approach to archeological osteological material is considerably more complex—particularly in regions such as Wolin, which functioned as dynamic contact zones and experienced substantial genetic influx. In such contexts, researchers attempt to reconstruct and classify fragmented remains into morphotypes using osteological methods, with the aim of drawing comparisons with modern breed types. Yet, as Hourani [123] emphasizes in his study of dog burials from Persian-period Beirut, such parallels should be approached with caution, as morphotypes do not always align clearly with modern breed classifications, especially in genetically diverse populations. The basic typological system distinguishes three principal skull forms: dolichocephalic (long-headed), mesocephalic (medium-headed), and brachycephalic (short-headed). While this tripartite model provides a useful framework, the application of cranial index values allows for more nuanced assessments of head shape. Such an analysis was carried out for the dogs from Wolin, with results presented in Table 10.

The data show that 50% of the analyzed skulls fall within the dolichocephalic category, while the remaining 50% are classified as sub-mesocephalic. For interpretive purposes, these morphotypes can be tentatively related to modern breed analogs—for example, the modern Doberman Pinscher exemplifies the dolichocephalic type, whereas breeds such as the Brittany Spaniel, the Cocker Spaniel, and the German Shorthaired Pointer represent the sub-mesocephalic type [123]. It should also be noted that within both dolichocephalic and mesocephalic categories, more extreme forms can be distinguished—referred to as ultradolichocephalic and ultramesocephalic, respectively. Osteological data thus suggest that early medieval *Canis familiaris* from Wolin were predominantly characterized by long- and medium-cranial forms.

Regarding body size, shoulder height was estimated based on the greatest length of five bones from the free portions of the thoracic and pelvic limbs, as well as the scapula, which represents the girdle segment of the thoracic limb. These calculations indicate that the Wolin dog population included both very small individuals (25–30 cm at the shoulder) and large ones (60–70 cm). This size variability, combined with observed cranial diversity, likely reflects the functional and social roles of dogs within the settlement, including guarding, herding, and hunting.

The skeletal features documented here not only confirm earlier anatomical conclusions about medieval Polish dogs but also expand them with new observations. A coordinated synthesis of data from West Pomerania can serve as a model comparative framework for future typological and metric studies of morphological variability and evolution in commensal animals. Such an integrated interpretation provides a coherent morphological “canvas” upon which individual site assemblages act as precise brushstrokes within a broader picture—allowing future researchers to build upon an organized database where each new find enriches our understanding of animal body-form dynamics across space and time.

## 5. Conclusions

The comprehensive osteometric analysis of 209 dog remains from early medieval Wolin, conducted in this study, revealed clear variation in skull and long bone dimensions. This indicates the presence of at least two—and likely three—main phenotypes of the species within the urban port settlement. Shoulder height, estimated from braincase length, scapulae, and long bones, ranged from approximately 40 cm to over 60 cm, with body mass estimates varying between 4 and 40 kg. The age distribution (3–10 years) and sex identification using the B100/L index confirm that both young and mature individuals of both sexes were actively utilized by the inhabitants of Wolin.

The observed morphological diversity suggests multifunctional roles for these dogs: smaller phenotypes likely served as guard companions or as small-game hunters, while larger ones may have been used for herding, transport, or as status symbols. The presence of worn teeth and cranial injuries may point to harsh living conditions or use in property protection. The diversity of dog morphotypes in Wolin is likely linked to international trade and the exchange of animals between different populations. While the osteometric results provide valuable data on bone size and proportions, they also come with limitations. Body mass estimates are based on calibration formulas that may not fully account for inter-population differences. They also do not reflect soft tissue variability or the influence of diet and disease. Furthermore, the state of preservation and mechanical damage to remains may affect measurement accuracy.

Taking these factors into account, the results not only enhance our understanding of early medieval dogs in the Baltic region but also underscore Wolin’s role as a center of cultural and biological diversity. Further research incorporating isotopic and genetic analyses will help test the proposed functional interpretations and deepen our understanding of domestication and morphological selection processes in the region. An open question remains: how did these morphotypes evolve over the following centuries AD?

## Figures and Tables

**Figure 2 animals-15-02171-f002:**
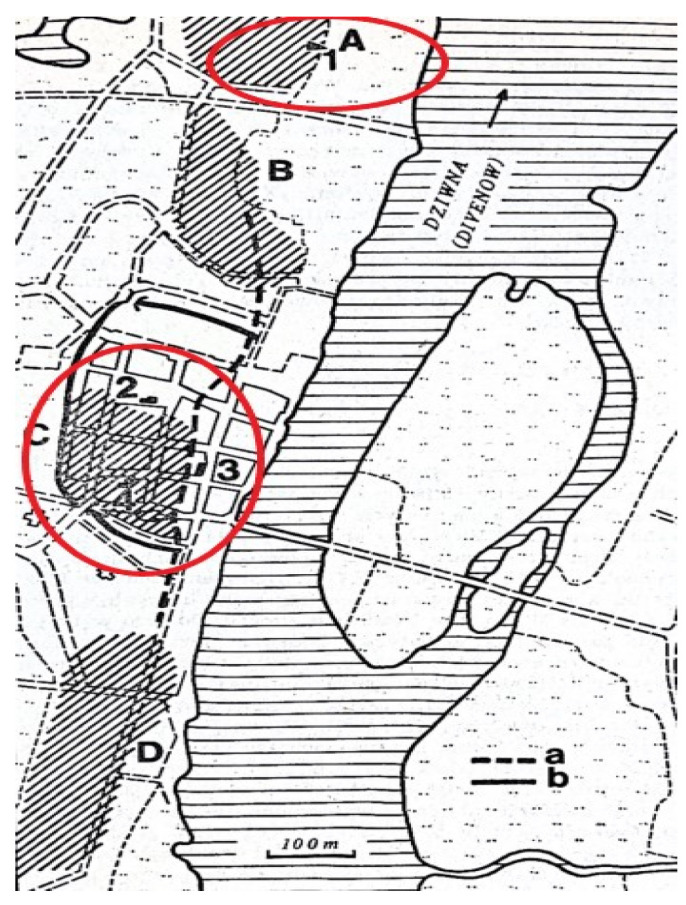
Locations of the excavation sites in Wolin. Capital letters indicate (A) Srebrne Wzgórze (Silver Hill) and (C) City Center; lowercase letters indicate (a) former coastline and (b) fortifications, modified after Ważny and Eckstein 1987, p. 148, [71].

**Figure 3 animals-15-02171-f003:**
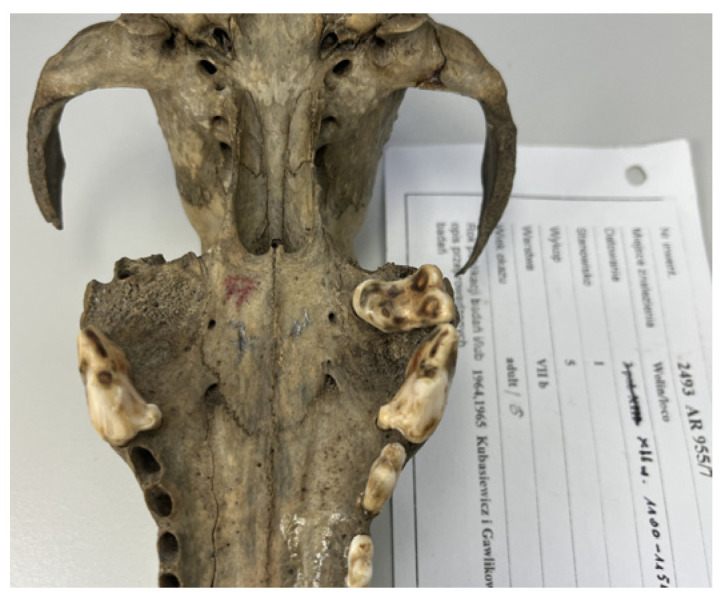
Ventral view of a canine skull showing an underdeveloped alveolus for the first molar (M^1^).

**Figure 4 animals-15-02171-f004:**
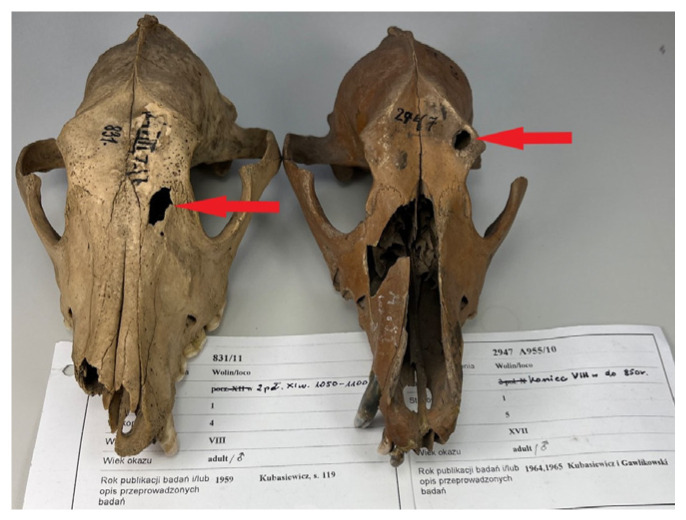
Evidence of bone defects at the junction between the neurocranium and the viscerocranium in a canine skull.

**Figure 5 animals-15-02171-f005:**
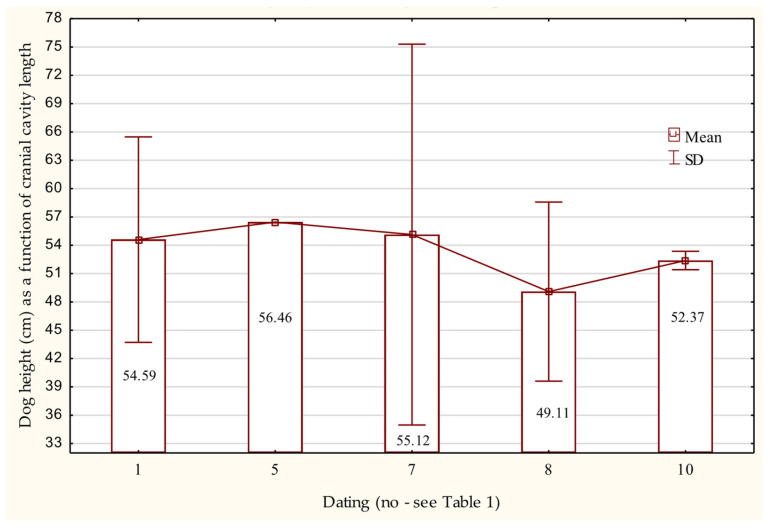
Withers height of dogs based on cranial cavity length in the Early Medieval period.

**Figure 6 animals-15-02171-f006:**
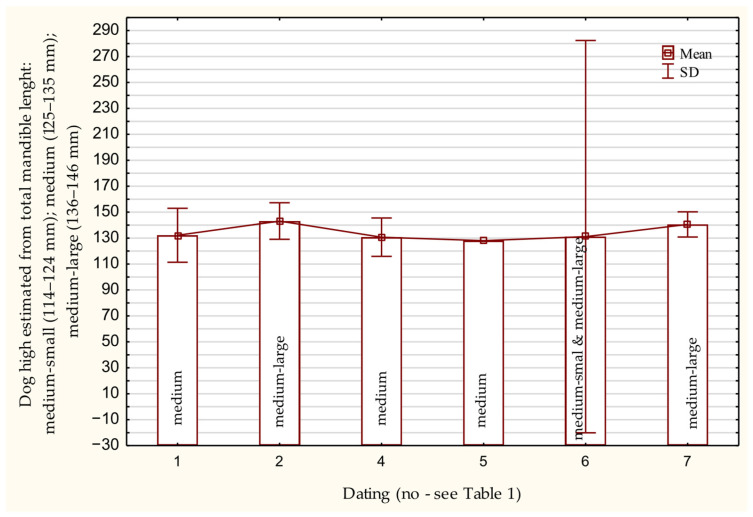
Height of the withers of the dog based on the length of the mandibula bone in the Early Middle Ages.

**Figure 7 animals-15-02171-f007:**
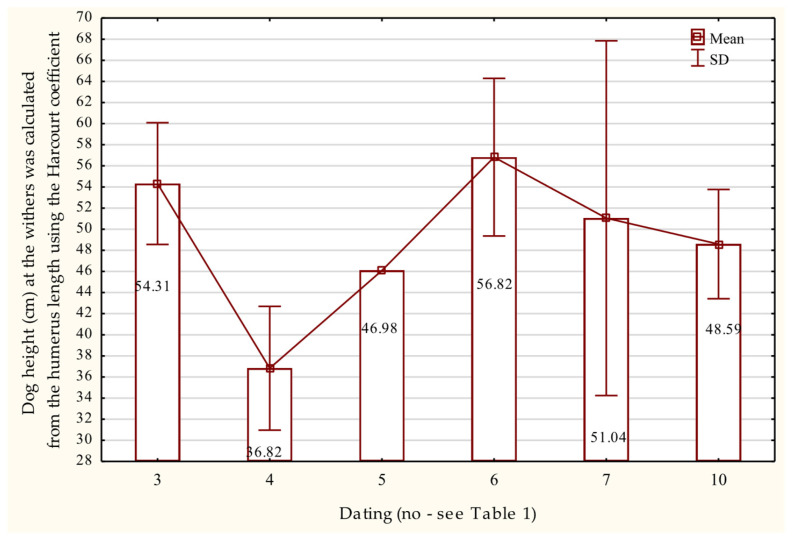
Height at the withers of dogs based on the length of the humerus bone in the Early Middle Ages.

**Figure 10 animals-15-02171-f010:**
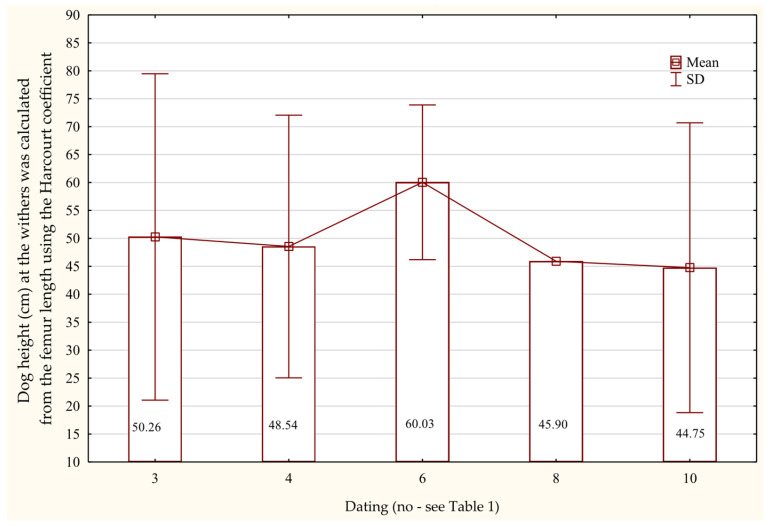
Height at the withers of dogs based on the length of the femur bone in the Early Middle Ages.

**Figure 11 animals-15-02171-f011:**
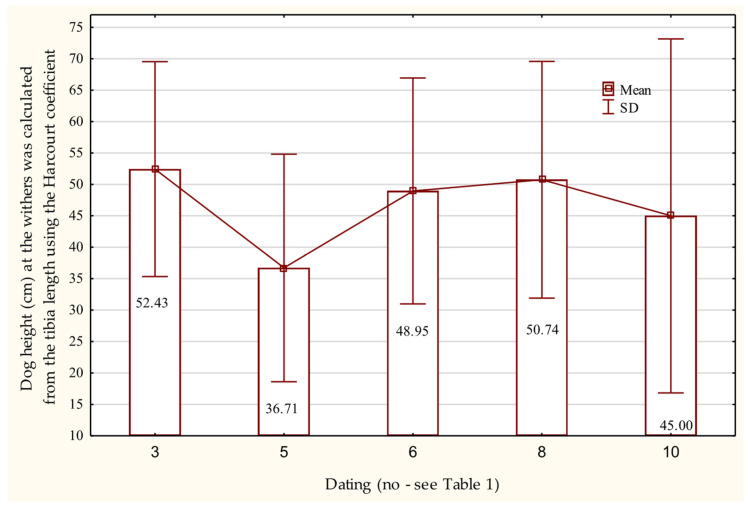
Height at the withers of dogs based on the length of the tibia bone in the Early Middle Ages.

**Table 1 animals-15-02171-t001:** Chronology and NISP (number of identified specimens) of dog remains from Wolin Island used in the study.

No.	Dating	Cranial Skeleton		Postcranial Skeleton						In Total	%
		C	M	S	H	R	U	F	T		
1.	The end of the 8th century to the year 850	3								3	1.44
2.	The second half of the 9th century AD		4							4	1.92
3.	From the second half of the 9th century to the turn of the 9th to 10th centuries AD		7	5	4	8		3	6	33	15.86
4.	From the end of the 9th century to around the middle of the 10th century AD		3		4		2	3		12	5.77
5.	From the middle of the 10th century to the end of the 3rd quarter of the 10th century AD	1	8	5	7	5	2		7	35	16.83
6.	The turn of the 10th to 11th centuries or the first half of the 11th century AD		4	6	4	2	5	4	3	28	13.46
7.	From the middle of the 11th century to the beginning of the 2nd half of the 11th century AD (1000–1050)	3	5		7					15	7.22
8.	The second half of the 11th century AD (1050–1100)	2				7	2	2	4	17	8.17
9.	From the end of the 11th century to the second half of the 12th century AD		10			3				13	6.25
10.	The second half of the 12th century (1100–1150) to the beginning of the 13th century AD	4	6	5	7	9	6	4	7	48	23.08
In total		13	47	21	33	34	17	16	27	208	100

Explanations in the table: C—skull; M—mandibula; S—scapula; H—humerus; R—radius; U—ulna; F—femur; T—tibia. The dog remains from Wolin Island were dated using a combined approach based on three complementary sources of evidence: (1) early work by Kubasiewicz [76]; (2) dendrochronological data published by Ważny and Eckstein [71] for the early medieval Slavic settlement of Wolin; and (3) a relative chronology developed at the Institute of Archaeology and Ethnology of the Polish Academy of Sciences in Szczecin, based on the proportional analysis of pottery types across successive occupation phases.

**Table 2 animals-15-02171-t002:** Sex prediction index for male and female dogs from the 8–13th centuries and the 21st century AD.

Index	Sex	Century					
		8–13th AD			21st AD		
		n	x ± s	Min–Max	n	x ± s	Min–Max
B * 100/L [90]	male	9	103.69 ± 11.79	82.31–116.58	9	101.79 ± 18.42	74.57–132.82
	female	4	143.78 ± 5.51	139.14–150.88	4	148.25 ± 9.09	137.18–157.92

**Table 3 animals-15-02171-t003:** Cranial measurements of dogs found on Wolin Island, dated to the late 8th century through the third quarter of the 13th century AD, based on the method by von den Driesch [92] (^♠^—(A-N)x(Eu-Eu)x(CH); *—not in v.d.D.).

Osteometrics (mm).		n	x ± s	Min	Max
1	Total length	8	197.25 ± 14.95	182.00	223.00
2	Condylobasal length	8	184.44 ± 14.17	171.00	209.00
3	Basal length	8	174.56 ± 19.49	162.00	192.00
4	Basicranial axis	12	47.44 ± 13.62	42.71	54.74
5	Basifacial axis	8	127.38 ± 9.81	115.75	144.16
6	Neurocranium length	10	104.40 ± 6.45	98.00	115.00
7	Upper neurocranium length	12	92.36 ± 8.06	79.89	109.04
8	Viscerocranium length	8	98.44 ± 9.46	89.88	116.83
9	Facial length	8	113.85 ± 9.38	103.48	129.30
10	Greatest length of the nasals	8	78.05 ± 7.56	72.74	93.26
11	Length of braincase (*Basion–Ethmoideum*)	13	83.44 ± 5.62	75.65	96.22
12	Snout length	8	89.56 ± 6.78	81.98	171.88
13	Median palate length	8	96.98 ± 7.02	89.64	107.51
13a	Palatal length	8	94.69 ± 6.07	88.33	103.45
14	Length of the horizontal part of the palatine	9	33.99 ± 3.11	29.04	38.12
15	Length of the cheektooth row	10	68.82 ± 4.03	62.70	75.63
16	Length of the molar row	10	21.02 ± 1.57	18.95	23.36
17	Length of the premolar row	10	53.15 ± 3.33	48.03	60.37
19	Length of the carnassial alveolus	10	18.01 ± 0.63	16.89	18.66
19a	Length of P^4^	10	18.14 ± 0.50	17.13	18.90
19b	Greatest breadth of P^4^	10	9.79 ± 1.30	7.24	11.74
19c	Breadth of P^4^	10	6.62 ± 1.02	5.29	8.26
20	Length of M^1^	7	12.94 ± 0.86	11.77	14.03
20a	Breadth of M^1^	7	16.67 ± 0.80	15.33	17.87
21	Length of M^2^	8	7.30 ± 0.46	6.74	8.01
21a	Breadth of M^2^	8	10.39 ± 0.54	9.72	11.40
22	Greatest diameter of auditory bulla	12	22.45 ± 2.95	19.00	27.06
23	Greatest mastoid breadth	11	65.59 ± 4.20	60.00	74.00
24	Breadth dorsal to the external auditory meatus	12	63.33 ± 4.77	56.45	72.72
25	Greatest breadth of the occipital condyles	13	36.75 ± 2.42	34.43	43.04
26	Greatest breadth of the bases of the paraoccipital processes	11	53.16 ± 5.61	46.28	62.68
27	Greatest breadth of the foramen magnum	12	18.56 ± 1.10	16.77	20.44
28	Height of the foramen magnum	13	16.24 ± 1.80	12.95	19.36
29	Greatest breadth of the braincase	13	56.85 ± 3.26	50.80	62.00
30	Zygomatic breadth	8	105.36 ± 9.12	90.60	121.82
31	Least breadth of the skull	13	38.45 ± 3.24	32.75	45.36
32	Frontal breadth	13	50.97 ± 7.28	42.07	68.01
33	Least breadth between orbits	10	39.68 ± 5.93	32.66	50.04
34	Greatest palatal breadth	9	61.70 ± 4.35	56.66	69.95
35	Least palatal breadth	9	34.05 ± 3.01	30.37	39.37
36	Breadth at the canine alveoli	7	35.41 ± 3.51	31.85	41.04
37	Greatest inner high of the orbit	10	31.67 ± 3.24	26.47	36.96
38	Skull height	13	53.17 ± 5.49	46.05	65.07
39	Skull height without the sagittal crest	13	46.19 ± 3.04	41.73	51.77
40	Height at the occipital triangle	13	40.87 ± 3.24	36.52	47.03
41	Height of the canine	5	15.19 ± 6.42	9.41	24.26
42 ^♠^	Neurocranium capacity	8	40.59 ± 7.37	32.72	51.63
43 *	Maxillofacial width	10	52.53 ± 6.88	42.17	64.56
44 *	Zygomatic length	12	95.71 ± 7.57	83.99	107.74
45 *	Foramen magnum area	12	198.53 ± 31.02	150.21	261.52
46 *	Occipital triangle area	11	1093.35 ± 144.26	905.93	1384.26

**Table 4 animals-15-02171-t004:** Cranial indexes of dog skulls found on Wolin Island, dated to the period from the late 8th century to the third quarter of the 13th century AD.

Index	Statistical Values			
	n	x ± s	Min	Max
Zy–Zy × 100/A–P	7	53.15 ± 2.90	48.76	58.16
Eu–Eu × 100/A–N	8	58.74 ± 3.85	54.75	64.88
Zy–Zy × 100/N–P	7	106.49 ± 8.54	95.13	117.78
Eu–Eu × 100/A–P	8	29.43 ± 1.68	27.80	32.48
Eu–Eu × 100/B–P	8	33.24 ± 1.90	31.01	36.48
Eu–Eu × 100/B–P	8	31.49 ± 2.16	29.16	34.88
N–B × 100/B–P	7	60.18 ± 1.66	57.99	62.50
Pm–Pd × 100/St–P	8	54.82 ± 3.04	49.94	58.74
(A–N) × (Eu–Eu) × (CH)	8	40.59 ± 7.37	32.77	51.63
B–S × 100/P–S	8	37.71 ± 2.74	33.66	43.29
Ect–Ect × 100/A–P	8	29.93 ± 2.14	24.43	30.50
N–A × 100/N–P	8	100.72 ± 5.72	90.87	107.93
Eu–Eu × 100/F(So)–O	12	61.51 ± 4.86	53.40	71.23
Palatal width × 100/Palatal length	8	63.83 ± 2.64	59.31	66.97
Canine width/Palatal length	7	36.92 ± 2.24	32.98	38.88
Area *Foramen magnum*/Area of *occipital triangle*	10	18.21 ± 2.54	15.44	22.77
Area of *occipital triangle*/Ot–Ot	11	16.63 ± 1.51	14.02	18.71
B–P/Area of *occipital triangle*	8	16.09 ± 1.67	13.87	18.27
P^1^–M^2^/A–P	8	35.10 ± 1.69	33.60	38.36
Wyrost–Kucharczyk Coefficient = (1.016 × B–E) − 31.2	13	53.58 ± 5.72	45.66	66.56
^♠^ (*Maxillofacial* width)^2^/B–P	8	15.93 ± 3.74	11.29	21.71
Forehead Width Index Ect–Ect/B–P	8	30.44 ± 2.84	27.61	35.42
Exponential Forehead Width Index (Ect–Ect)^2^/B–P	8	16.44 ± 4.23	12.89	24.09
Projected Viscerocranial Length (P–Ect) × 100/B–P	8	65.90 ± 1.25	63.36	67.45
Orbital Position Index (P–Ect × 100/A–Ect)	8	121.28 ± 3.98	116.13	128.46
Braincase Length Index (A–Ect) × 100/B–P	8	54.38 ± 1.85	52.06	56.52
^♥^ Forehead Position Index P–So × 100/So–O	8	120.47 ± 8.83	104.25	129.92
Tooth Length Index P^1^–M^2^/B–P	8	39.15 ± 0.96	37.73	40.76
Exponential Tooth Length Index (P^1^–M^2^)^2^/B–P	8	26.77 ± 2.08	23.49	29.83
^♦^ Index Zygomatic Arch Length/A–Zy	9	1.13 ± 0.07	1.02	1.25
Index Facial Skull Length N–Rh/N–P	5	0.70 ± 0.04	0.67	0.78
Index1 Eu–Eu/N–Rh	8	0.75 ± 0.05	0.66	0.82
Index ‘1 Eu–Eu/N–P	8	0.59 ± 0.04	0.53	0.66
Index2 P–A/Zy–Zy	7	1.89 ± 0.14	1.72	2.05
Index ‘2 P–B/Zy–Zy	7	1.97 ± 0.13	1.53	1.85
Index3 A–N/Zy–Zy	7	0.95 ± 0.08	0.87	1.05
Index4 A–N/Eu–Eu	8	1.71 ± 0.11	1.54	1.83

Explanation: ^♠^ Facial width (maxillofacial width) was estimated by measuring the distance between the lowest points of the zygomaticomaxillary sutures on both sides of the maxilla (*sutura zygomaticomaxillaris sinistra* and *dextra*). ^♥^ Forehead position index = (P–So) × 100/(So–O), where So (*Sagectorbion*) is the point of intersection between the sagittal suture and the line connecting the most lateral points of the zygomatic supraorbital processes of the frontal bones. ^♦^ Zygomatic arch length was measured from the *Otion* to the most rostral point of the zygomaticomaxillary suture.

**Table 5 animals-15-02171-t005:** Mandibular measurement of dogs found on Wolin Island, dated to the period from the late 8th century to the third quarter of the 13th century AD, based on the method by von den Driesch [92].

Osteometrics (mm)		Statistical Values			
		n	x ± s	Min	Max
Measurements of the mandible					
1	Total length	21	136.20 ± 11.08	113.50	150.00
2	Length: the angular process	12	140.39 ± 6.90	128.00	149.60
3	Length from the indentation between the condyle process and the angular process	19	131.84 ± 8.90	113.00	143.00
4	Length: the condyle process–the aboral border of the canine alveolus	25	121.05 ± 10.19	97.80	133.00
5	Length from the indentation between the condyle process angular process–the aboral border of the canine alveolus	23	116.30 ± 8.16	97.10	126.70
6	Length: the angular process–the aboral border of the canine alveolus	13	124.96 ± 6.83	112.50	134.20
7	Length: the aboral border of the alveolus of M_3_–the aboral of the canine alveolus	29	76.92 ± 8.90	52.40	86.70
8	Length of the cheektooth row, M_3_-P_1_ *	27	71.77 ± 8.35	49.10	83.20
9	Length of the cheektooth row, M_3_-P_2_ *	27	68.80 ± 5.09	58.20	77.50
10	Length of the molar row	31	34.46 ± 3.50	25.10	38.80
11	Length of the premolar row, P_1_–P_4_	28	39.79 ± 3.13	33.60	45.00
12	Length of the premolar row, P_2_–P_4_	30	34.66 ± 2.72	28.70	39.30
13	Length of the carnassial	24	21.15 ± 1.54	16.50	23.50
13a	Breadth of the carnassial	24	8.85 ± 0.65	7.40	10.00
14	Length of the carnassial alveolus	32	20.09 ± 1.84	15.80	22.80
15	Length of M_2_	17	8.61 ± 0.99	6.00	10.00
15a	Breadth of M_2_	17	6.49 ± 0.71	4.90	7.70
16	Length of M_3_ and breadth of M_3_	teeth absent			
17	Greatest thickness of the body of the jaw	31	11.02 ± 1.17	8.20	13.30
18	Height of the vertical ramus: basal point of the angular process	13	54.68 ± 2.94	50.00	59.50
19	Height of the mandible behind M_1_	31	22.57 ± 2.48	15.50	27.20
20	Height of the mandible behind P_2_ and P_3_	32	18.55 ± 1.78	12.80	21.30
21	Height of the canine	10	19.97 ± 8.82	9.77	42.00

Explanation: *—measured along the alveoli.

**Table 6 animals-15-02171-t006:** Mandibular indices of dog skulls found on Wolin Island, dated from the late 8th century to the third quarter of the 13th century AD.

Index	Statistical Values			
	n	x ± s	Min	Max
Height of the vertical ramus * 100/total length	12	38.89 ± 1.95	34.97	42.22
Height of the mandible behind M_1_ * 100/total length	21	16.64 ± 0.92	15.10	18.44
Height of the vertical ramus * 100/length of the cheektooth row M_3_-P_1_	11	73.77 ± 3.71	65.53	78.44
Height of the mandible behind M_1_ * 100/length of the molar row	29	64.85 ± 3.70	59.21	74.32
Greatest thickness of the body of jaw * 100/total length	21	8.08 ± 0.57	6.94	9.38
Greatest thickness of the body of jaw * 100/height of the mandible behind M_1_	29	49.03 ± 3.36	41.20	55.75

**Table 7 animals-15-02171-t007:** Postcranial measurements of dogs found on Wolin Island, dated to the late 8th century through the third quarter of the 13th century AD, based on von den Driesch [92].

Osteometrics (mm)	n	x ± s	Min	Max
Scapula				
Height	5	126.94 ± 20.05	92.70	144.00
Diagonal height	1	129.79	-	-
Greatest dorsal length	1	67.62	-	-
Smallest length of the neck of the scapula	19	24.88 ± 3.69	13.90	30.05
Greatest length of the glenoid process	19	30.36 ± 3.97	18.70	37.40
Length of the glenoid cavity	18	27.72 ± 3.35	17.20	30.00
Breadth of the glenoid cavity	16	17.95 ± 1.78	14.00	20.50
Humerus				
Greatest length	15	152.31 ± 28.33	100.00	188.80
Greatest length from the head (caput	13	150.29 ± 25.87	97.20	181.30
Greatest depth of the proximal end	14	29.00 ± 5.86	19.60	39.40
Depth of the proximal end	14	38.46 ± 6.62	27.20	47.00
Smallest breadth of the diaphysis	29	12.64 ± 1.76	8.80	15.50
Greatest breadth of the distal end	30	31.50 ± 4.23	21.70	37.60
Greatest breadth of the trochlea	28	22.65 ± 3.55	14.80	33.68
Circumference of the humerus measured at a distance of 35% from the distal end of its diaphysis	28	41.17 ± 5.60	27.50	52.00
Radius				
Greatest length	19	146.31 ± 37.40	92.50	203.00
Greatest breadth of the proximal end	32	17.57 ± 1.90	12.80	21.50
Smallest breadth of the diaphysis	31	12.83 ± 1.73	9.00	15.40
Smallest circumference of the diaphysis	31	34.07 ± 4.01	25.50	40.50
Breadth of the distal end	20	23.25 ± 3.90	12.60	28.00
Ulna				
Greatest length	12	146.34 ± 34.56	96.30	221.00
Depth across the processus anconaeus	17	23.05 ± 4.09	15.80	29.08
Smallest depth of the olecranon	17	19.42 ± 4.03	11.80	26.89
Greatest breadth across the coronoid process	17	15.30 ± 2.28	11.50	19.20
Femur				
Greatest length	13	165.01 ± 36.07	99.00	214.00
Greatest length from the caput femoris	10	162.07 ± 43.23	93.50	214.00
Greatest breadth of the proximal end	14	36.59 ± 5.92	25.00	44.03
Greatest depth of the caput femoris	16	18.00 ± 2.76	12.10	20.80
Smallest breadth of the diaphysis	16	12.33 ± 1.38	9.40	14.40
Femoral circumference taken at the midpoint on the long axis	16	40.30 ± 5.41	28.00	47.50
Greatest breadth of the distal end	12	31.15 ± 5.64	21.80	41.70
Tibia				
Greatest length	21	158.93 ± 36.68	93.00	210.00
Greatest breadth of the proximal end	15	32.51 ± 5.38	23.60	42.70
Smallest breadth of the diaphysis	24	12.38 ± 1.73	8.60	15.00
Greatest breadth of the distal end	20	21.31 ± 3.21	15.20	27.20

**Table 9 animals-15-02171-t009:** Estimated ranges of shoulder height (cm) of dogs from selected early medieval archeological sites in Pomerania, based on the maximum lengths of shoulder blades and long bones, calculated using the method by Harcourt [25] and Lasota-Moskalewska [93].

Bone	Early Medieval Archeological Site in the Polish Part of Pomerania			
	Wolin, Current Research	Kołobrzeg Kołobrzeg [81]	Dobra Nowogardzka [110]	Szczecin- Vegetable Market Szczecin [111]
Scapula	37.63–58.46	-	45.18	38.16–56.00
Humerus	31.64–63.62	47.42–50.88	–	–
Radius	29.78–65.50	37.99–39.47	30.0–54.0	62.76–65.04
Femur	29.79–65.90	-	57.5	61.40–62.84
Tibia	27.15–61.32	-	29.4	61.90–62.84

**Table 10 animals-15-02171-t010:** Typology of dog skulls from Wolin Island (late 8th century to the third quarter of the 13th century AD) based on cranial, viscerocranial, and neurocranial indices.

Skull Number	^(1)^ Zy–Zy/A–P × 100	^(2)^ N–A/N–P	^(3)^ Eu–Eu/N–A × 100	Skull Type per [123]	Dating (See Table 1)
A954	48.76	1.05	54.75	d	1
2947 A955	55.26	0.94	64.88	s-m	1
2572 A954	56.66	0.99	57.84	s-m	1
2948	-	-	-	-	5
437	49.84	0.91	58.40	d	7
439	-	-	-	-	7
2768 A955	-	-	-	-	7
829.830	48.97	1.03	56.32	d	8
831	54.40	1.08	55.04	s-d	8
2493 A955	58.16	1.02	64.18	s-m	10
2704 A995	-	1.03	58.49	d ^(3)^	10
2490	-	-	-	-	10
2595	-	-	-	-	10

Explanation: Abbreviations: d—dolichocephalic; s-d—subdolichocephalic; s-m—submesocephalic. ^(1), (3)^ According to Evans [124], the cranial index in dolichocephalic dogs should be close to 50. ^(2)^ According to Sisson and Grossman [125], the maximum cranial index for dolichocephalic dogs is approximately 50, while for mesaticephalic dogs it is around 70. A comparison of skull length to facial length expressed as an index value of approximately 3 indicates a brachycephalic type, whereas a value close to 2 indicates a mesaticephalic type. In the absence of a defined index value for the dolichocephalic type, a value approaching 1 may be assumed. ^(3)^ As for the cranial index, values around 56 are indicative of a mesaticephalic morphotype.

## Data Availability

The data presented in this study are available at the Department of Animal Anatomy and Zoology, Faculty of Biotechnology and Animal Husbandry, West Pomeranian University of Technology in Szczecin, Poland. Data are available upon request from the corresponding author.

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
