# Peer review of "Morphological and Metric Analysis of Medieval Dog Remains from Wolin, Poland"

_animals, 2025, doi:10.3390/ani15152171_

Round 1
Reviewer 1 Report
Comments and Suggestions for Authors
Comments to Author,
First of all, I would like to point out that this study is a valuable contribution to zooarchaeological studies on dog identification. In particular, a detailed evaluation of Wolin dogs has been carried out, revealing both morphological and morphometric data. For this reason, I consider this study important and approve it. I am also very pleased to have received this study. I believe that the revisions listed below will further enhance the overall value of the article. I recommend its publication after the necessary corrections have been made. However, I have some comments on certain issues, which I have listed below.
Page 2; lines 43-50:
This section should be enriched with literature on the relationship between humans and dogs from prehistoric times to the Middle Ages. Brief references to topics such as the purposes for which humans used dogs, the emergence of dog breeds, and changes observed in their morphology would be beneficial. A few sentences about the changes and relationship from the observation of Rutimeyer's Turbary dog in European Neolithic sites to the Middle Ages would improve the quality of the article. For this purpose, I thought it would be useful to recommend some literature.
Bökönyi, S., 1974. History of Domestic Mammals in Central and Eastern Europe. Akadémiai Kiadó, Budapest.
Schoenebeck, J.J., Hutchinson, S.A., Byers, A., Beale, H.C., Carrington, B., Faden, D.L., Rimbault, M., Decker, B., Kidd, J.M., Sood, R., Boyko, A.R., Fondon III, J.W., Wayne, R.K., Bustamante, C.D. Ciruna, B., Ostrander, E.A., 2012. Variation of BMP3 Contributes to Dog Breed Skull Diversity. PloS Genetics, 8 (8), e1002849. doi:10.1371/journal.pgen.1002849
Zedda, M., Manca, P., Chisu, V., Gadau, S., Lepore, G., Genovese, A., Farina, V., 2006. Ancient Pompeian Dogs – Morphological and Morphometric Evidence for Different Canine Populations. Anat. Histol. Embryol., 35, 319-324.
Rütimeyer, L., 1861. Die Fauna der Pfahlbauten in der Schweiz. Neue Denkschr. d. Allg. Schweiz. Ges. d. ges. Natwiss., 19, Basel.
Page 3; Lines 84-100.
Previous Research in Wolin: This section should focus on dog studies conducted in Wolin. It should be shortened considerably and combined with the Research Gap. It should also be filled with very brief information about the history of Wolin. The aim is to focus on dog remains.
Page 3-4; Lines 116-131 Outline of Wolin's History This section should be removed. It should be mentioned very briefly in the previous section.
Page 4,; Lines 133-139 Urban and Economic Development: This section should be removed.
Lines 140-146 Decline: This section should be removed.
Lines 147-152 History of Archaeological Research: This section should be removed.
Lines 153-167 Wolin Island – Location and Depositional Conditions: This section can be briefly mentioned in the history of Wolin.
Page 5; Line 183: The information provided in this section is quite valuable. If this section is included, the section titled ‘The Human–Dog Relationship in Prehistory and the Early Medieval Period’ can be removed.
Page 6; Line 209: Skull should be used instead of calvaria. This spelling should be used throughout the text and tables.
Page 8; lines 227-239: This paragraph should be moved to the findings section.
Page 8; line 243: ‘A Guide to the Measurement of Animal Bones from Archaeological Sites’ should be replaced with the author's name. For example, von den Driesch, 1976
Page 23; Line 539: What are the typological differences in the skulls of Wolin dogs? How many of these dogs are dolicho- and mesocephalic? Were any brachycephalic dogs found among the dog skulls?
Author Response
Dear Sirs,
I would like to inform that I have revised the manuscript Animals (ID: 3729714) in accordance with the reviewers’ suggestions.
I would also like to express my sincere gratitude to the reviewers for their valuable and constructive comments.
Below is a summary of the revisions and additions made:
In response to Reviewer 1 comments (referenced by line numbers):
Lines 43–50: This section has been reorganized and updated with the recommended literature, along with several additional sources. Thank you very much for these helpful suggestions.
Lines 84–100: The section has been revised for improved clarity and coherence, including the merging of selected paragraphs.
Lines 116–131: The reviewer’s comments have been fully addressed.
Page 4: The specified subsections have been removed, and the text has been revised accordingly.
Pages 5–6: All comments related to these pages have been taken into account.
Page 8: The paragraph in question has been moved to the Results section.
Line 243: The suggested correction has been implemented.
Page 23 – Comment regarding skull typology:
- a) No brachycephalic skulls were identified among the analyzed specimens.
- b) In response to the reviewer’s suggestion, an additional analysis was conducted. The results are presented in Table 10 and discussed in the Discussion section.
Taphonomic analysis
I would like to clarify that a classical taphonomic analysis of the dog bone material from Wolin was not undertaken, as the specimens originated from research and teaching collections that had been repeatedly relocated, used for instructional and exhibition purposes, and stored under varying conditions.
These factors led to secondary alterations of the bone surfaces (e.g., scratches, discoloration, microerosion), which could have obscured original taphonomic features.
For this reason, I deliberately refrained from recording taphonomic traces, recognizing that the available material did not provide a reliable basis for interpretation.
I would also like to emphasize that, unlike well-documented contexts—such as the puppy burial from San Nicolas Island (Vellanoweth et al. 2008)—the Wolin assemblage lacked evidence of its original depositional context.
Nevertheless, in my analysis I took into account and interpreted potentially traumatic lesions observed in the orbital regions of two skulls (nos. 831 and 2947), locations typically associated with perimortem impacts used to subdue dogs.
Such injuries have been described by, among others, Baker and Brothwell (1980) as traces of human intervention.
In conclusion, while a standard taphonomic approach was not feasible, all observations potentially indicative of deliberate human activity were considered. However, the primary research focus was on the morphology of the skulls and postcranial elements, which justifies limiting the taphonomic discussion to a few significant instances.
Clarification Regarding the Dating of Dog Bone Material from Wolin.
In response to the reviewer’s request concerning the dating method and the laboratory responsible for determining the chronology of the dog bone material from Wolin, I would like to clarify that a synthetic, indirect approach was employed in this study. Instead of direct radiocarbon dating of the osteological samples, the chronology was established based on an integrated analysis of stratigraphic data, archaeological context, and typological comparisons.
The inability to provide a specific laboratory name or analysis reference stems from the archival nature of the collection and the methodological practices employed during the earlier phases of research—prior to the widespread application of AMS radiocarbon dating in archaeozoological studies.
The adopted chronology rests on three principal foundations:
1.Early stratigraphic and archaeozoological research conducted in Wolin, particularly the pioneering work of M. Kubasiewicz (1959), along with subsequent publications by Kubasiewicz, Gawlikowski, and Stępień (1964, 1965, 1986, 1998). These studies identified excavation layers attributed to specific early medieval phases, which later served as a comparative framework for ongoing research.
2.Dendrochronological dating of wooden structures from Wolin, carried out by T. Ważny and D. Eckstein (1987) as part of a collaboration between the Institute of Wood Biology at the University of Hamburg and the Academy of Fine Arts in Warsaw. While these results pertain to architectural remains rather than directly to the bone material, they offer an important chronological reference for understanding the broader context of site occupation.
3.Chronological correlations based on early medieval pottery typologies, developed at the Institute of Archaeology and Ethnology of the Polish Academy of Sciences in Szczecin. The classification system introduced by Łosiński and Rogosz (1986a; 1986b), drawing on an earlier typology developed for Mecklenburg (Schuldt 1956), gained wide acceptance and was applied to numerous archaeological sites across the region. Based on statistical analysis of ceramic forms from stratified layers and their correlation with regional assemblages, this system enabled the dating of contexts from the second half of the 8th century to the 13th century.
Consequently, no direct radiocarbon analyses were performed on the examined bone material, and it is not possible to provide a laboratory name or analysis reference. Nonetheless, the proposed chronology is based on multiple lines of archaeological and environmental evidence, which together provide a sufficiently robust temporal framework for archaeozoological interpretation.
Reviewer 2 Report
Comments and Suggestions for Authors
The aims of this study are the following:
- Morphological and metrical analysis of the dog bone remains from the two sites of Early Medieval Wolin: Wolin Town and Silver Hill.
- Metrical analysis of the cranial and postcranial bone remains in order to infer the withers’ height and body mass;
The new data that include this research is the contextual interpretation of the functional roles of the dogs within the city’s social and economic structures in the Wolin community.
Please, include a map with the two sites.
Regarding the methodology, the table 1 indicates the NISP (identified number of specimen) of the dog bone remains. The caption in the table could be 'NISP per level' or a similar phrase.
The method of dating is necessary with the Laboratory name. Please, include it in a table.
Could be possible include some pictures of the dog bone remains? For example, remains with pathologies…
Have you found any taphonomic evidence on the bone surfaces?
Line 41. I suppose that the author means the dog skull morphology. Please clarity it, because Canis includes Canis lupus and Canis familiaris.
Line 176: circa is in italics
Please check the citation format according to the journal's guidelines.
The conclusions are consistent with the evidence and arguments and the references are appropriate.
Author Response
Dear Sirs,
I would like to inform that I have revised the manuscript Animals (ID: 3729714) in accordance with the reviewers’ suggestions.
I would also like to express my sincere gratitude to the reviewers for their valuable and constructive comments.
Below is a summary of the revisions and additions made:
In response to Reviewer 2 comments:
Maps showing the location of Wolin at the mouth of the Oder River on the southern Baltic coast, as well as the locations of the Srebrne Wzgórze and Wolin-Miasto archaeological sites (from which the studied remains were recovered), are now included as Figures 1 and 2. I confirm that written permission has been obtained from the publisher of Materiał Zachodniopomorskie for the use of the figures labeled as Figure 1 and Figure 2 in this manuscript.
Photographs of selected dog skulls have been included as Figures 3 and 4.
The reviewer’s question regarding taphonomic evidence is addressed in a dedicated section.
The dating method and the issue of laboratory attribution have also been clarified in the revised version of the manuscript.
Please do not hesitate to contact me should you require any further information.
Taphonomic analysis
I would like to clarify that a classical taphonomic analysis of the dog bone material from Wolin was not undertaken, as the specimens originated from research and teaching collections that had been repeatedly relocated, used for instructional and exhibition purposes, and stored under varying conditions.
These factors led to secondary alterations of the bone surfaces (e.g., scratches, discoloration, microerosion), which could have obscured original taphonomic features.
For this reason, I deliberately refrained from recording taphonomic traces, recognizing that the available material did not provide a reliable basis for interpretation.
I would also like to emphasize that, unlike well-documented contexts—such as the puppy burial from San Nicolas Island (Vellanoweth et al. 2008)—the Wolin assemblage lacked evidence of its original depositional context.
Nevertheless, in my analysis I took into account and interpreted potentially traumatic lesions observed in the orbital regions of two skulls (nos. 831 and 2947), locations typically associated with perimortem impacts used to subdue dogs.
Such injuries have been described by, among others, Baker and Brothwell (1980) as traces of human intervention.
In conclusion, while a standard taphonomic approach was not feasible, all observations potentially indicative of deliberate human activity were considered. However, the primary research focus was on the morphology of the skulls and postcranial elements, which justifies limiting the taphonomic discussion to a few significant instances.
Clarification Regarding the Dating of Dog Bone Material from Wolin.
In response to the reviewer’s request concerning the dating method and the laboratory responsible for determining the chronology of the dog bone material from Wolin, I would like to clarify that a synthetic, indirect approach was employed in this study. Instead of direct radiocarbon dating of the osteological samples, the chronology was established based on an integrated analysis of stratigraphic data, archaeological context, and typological comparisons.
The inability to provide a specific laboratory name or analysis reference stems from the archival nature of the collection and the methodological practices employed during the earlier phases of research—prior to the widespread application of AMS radiocarbon dating in archaeozoological studies.
The adopted chronology rests on three principal foundations:
1.Early stratigraphic and archaeozoological research conducted in Wolin, particularly the pioneering work of M. Kubasiewicz (1959), along with subsequent publications by Kubasiewicz, Gawlikowski, and Stępień (1964, 1965, 1986, 1998). These studies identified excavation layers attributed to specific early medieval phases, which later served as a comparative framework for ongoing research.
2.Dendrochronological dating of wooden structures from Wolin, carried out by T. Ważny and D. Eckstein (1987) as part of a collaboration between the Institute of Wood Biology at the University of Hamburg and the Academy of Fine Arts in Warsaw. While these results pertain to architectural remains rather than directly to the bone material, they offer an important chronological reference for understanding the broader context of site occupation.
3.Chronological correlations based on early medieval pottery typologies, developed at the Institute of Archaeology and Ethnology of the Polish Academy of Sciences in Szczecin. The classification system introduced by Łosiński and Rogosz (1986a; 1986b), drawing on an earlier typology developed for Mecklenburg (Schuldt 1956), gained wide acceptance and was applied to numerous archaeological sites across the region. Based on statistical analysis of ceramic forms from stratified layers and their correlation with regional assemblages, this system enabled the dating of contexts from the second half of the 8th century to the 13th century.
Consequently, no direct radiocarbon analyses were performed on the examined bone material, and it is not possible to provide a laboratory name or analysis reference. Nonetheless, the proposed chronology is based on multiple lines of archaeological and environmental evidence, which together provide a sufficiently robust temporal framework for archaeozoological interpretation.
Round 2
Reviewer 1 Report
Comments and Suggestions for Authors
I have reviewed the revised version of the article and the answers and corrections made by the author(s) are appropriate for the quality of the article. I find it appropriate to publish it in this form.